# GMSA: Enhancing Context Compression via Group Merging and Layer Semantic Alignment

## Abstract

Large Language Models (LLMs) have achieved impressive performance in a wide range of Natural Language Processing (NLP) tasks. However, when applied to long-context scenarios, they face two challenges, *i.e.*, computational inefficiency and redundant information. This paper introduces **GMSA**, a context compression method based on the encoder-decoder architecture, addressing these challenges by reducing input sequence length and redundant information. Structurally, GMSA has two key components: **Group Merging** and **Layer Semantic Alignment (LSA)**. Group merging is used to extract summary vectors evenly and efficiently from the original context. Layer semantic alignment, on the other hand, aligns the high-level abstract summary vectors with the low-level primary input semantics, thus bridging the semantic gap between different layers. In the training process, GMSA first learns soft tokens that contain nearly complete semantics via autoencoder training. To further adapt GMSA to downstream tasks, we propose **Knowledge Extraction Fine-tuning (KEFT)** to extract task-relevant knowledge from these soft tokens. GMSA not only significantly outperforms the traditional compression paradigm in context restoration but also achieves stable and significantly faster convergence with only a few encoder layers. We further evaluate GMSA on question-answering, summarization, and general knowledge retention capabilities across two backbones (*i.e.*, LLaMA-2-7B and Qwen2-7B), demonstrating its effectiveness and superiority, *e.g.*, on the NaturalQuestions dataset, GMSA can achieve approximately a 2x speedup in end-to-end inference while outperforming various methods by a large margin.[1]

## 1 Introduction

Thanks to powerful reasoning and generalization capabilities, Large Language Models (LLMs) have achieved remarkable performance across various Natural Language Processing (NLP) tasks (Touvron et al., 2023; Team et al., 2025; DeepSeek-AI et al., 2025; Qwen et al., 2025). However, directly applying LLMs to long-context scenarios presents two challenges: (1) Computational inefficiency. When processing long prompts, the quadratic complexity of the Transformer's attention mechanism (Vaswani et al., 2017) results in long inference latency. (2) Redundant information. Much redundant information in long-context scenarios can degrade model performance (Jiang et al., 2024).

Prompt compression methods address these two challenges by significantly reducing input length and removing redundant information. Prompt compression can be categorized into hard prompt compression (Li et al., 2023; Jiang et al., 2023; Pan et al., 2024; Jiang et al., 2024; Tang et al., 2025; Zhou et al., 2025; Cao et al., 2025; Chen et al., 2025; Zhao et al., 2025) and soft prompt compression (Mu et al., 2023; Chevalier et al., 2023; Ge et al., 2024; Zhang et al., 2024; **?**). Hard prompt compression involves deleting certain tokens from the original context to achieve compression. However, this explicit compression approach inevitably compromises semantic integrity. In contrast, leveraging the inherent redundancy in high-dimensional vector data, soft prompt compression learns a set of soft tokens with a length much shorter than the original context, enabling compression while preserving nearly complete semantics.

---

[1]Core code implementing GMSA and the baselines is provided in the supplementary material.

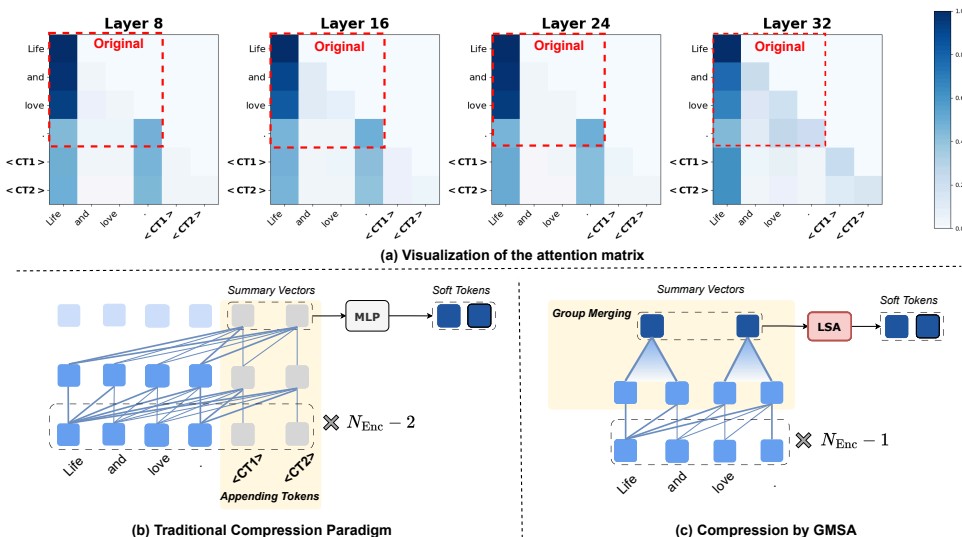

Figure 1: Traditional Compression Paradigm vs. Compression by GMSA. (a) visualizes the attention matrix when processing "Life and love. <CT1> <CT2>", where "<CT1>" and "<CT2>" are randomly initialized tokens. "Original" shows the attention changes during processing of "life and love." across different layers. (b) refers to the traditional compression paradigm. It first learns summary vectors in an autoregressive manner layer by layer, and then employs coarse-grained semantic alignment via a Multi-Layer Perceptron (MLP), where $N_{\mathrm{Enc}}$ is the number of encoder layers. (c) denotes the compression paradigm of GMSA, which first learns summary vectors via group merging and achieves semantic alignment between different layers via the Layer Semantic Alignment (LSA) module.

Although existing soft prompt compression methods can effectively reduce the number of input tokens, they have two limitations: (1) Ueven semantic learning in autoencoder training. Soft prompt compression typically relies on autoencoder-based training to ensure that the compressed representations retain as complete semantic information as possible (Ge et al., 2024; Cheng et al., 2024; Li et al., 2025; Liao et al., 2025; Dai et al., 2025; Rau et al., 2025; Choi et al., 2025). As shown in Figure 1, the compression process in traditional paradigm learns summary vectors layer by layer via appending learnable tokens. During this process, LLM tends to aggregate information on a few anchor tokens (Zhang et al., 2023; Xiao et al., 2023; Wang et al., 2023; Huang et al., 2024). Consequently, the semantics of anchor tokens (*i.e.*, "Life" and ".") are emphasized layer by layer, resulting in the semantics of the summary vectors being dominated by them while the semantics of other tokens are relatively diluted (*i.e.*, uneven semantic learning). Because the compressed representation overly depends on only a few tokens, fine-grained semantic details from the original context struggle to be fully preserved, thereby increasing the difficulty for the autoencoder training (*i.e.*, struggling to reconstruct the original context) (see Appendices B, J, and K for detailed empirical validation); (2) Ignoring the large semantic gap between different layers in the LLMs (Liu et al., 2024b; Jin et al., 2025). The summary vectors, which represent high-level abstract semantics, are directly treated as ordinary tokens (*i.e.*, as low-level semantic information) and fed into the decoder during training and testing, resulting in a large semantic gap. Therefore, two research questions naturally arise: (1) *How can we learn semantics more evenly and efficiently?* (2) *How can we bridge the large semantic gap between different layers?*

To this end, we propose **GMSA** (Context Compression via **G**roup **M**erging and Layer **S**emantic **A**lignment), a context compression framework based on the encoder-decoder architecture, which addresses these limitations from a structural perspective. Specifically, we tackle the first limitation through **Group Merging**. Group merging performs grouping and merging operations on the last hidden state of the encoder (see Figure 1). In particular, Group merging treats each group equally and merges all tokens within each group via averaging pooling, thereby avoiding information dilution and enabling more evenly semantic learning. *This step not only helps preserve more complete semantic information and is highly efficient, achieving more evenly and efficient semantic learning.*

Subsequently, to address the second limitation, we bridge the gap between high-level abstract semantic information and low-level primary input semantics by passing the summary vectors through the **Layer Semantic Alignment (LSA)** module, which is composed of a few Transformer blocks initialized with the weights of lower-layer decoder blocks (see Figure 2). *This step allows the summary vectors containing high-level abstract semantic information to be mapped into a low-level semantic space, thereby bridging the large semantic gap between different layers.*

During the training process, GMSA first employs the autoencoder training to ensure that the generated soft tokens contain nearly complete semantic information. Building on this foundation, we further propose **Knowledge Extraction Fine-tuning (KEFT)** to adapt GMSA to downstream tasks. Specifically, we freeze the encoder and LSA (which, after autoencoder training, can already produce soft tokens containing nearly complete semantics) and fine-tune the decoder to enhance its ability to extract task-relevant knowledge from the soft tokens.

Our contributions are threefold: (1) Structurally, we introduce the GMSA, which evenly and efficiently learns summary vectors through group merging and bridges the semantic gap between different layers via a Layer Semantic Alignment (LSA) module; (2) In the training process, we propose Knowledge Extraction Fine-tuning (KEFT) to guide the decoder to extract the knowledge required by downstream tasks from soft tokens; (3) Experimental results on diverse tasks (*e.g.*, QA, summarization, general knowledge retention) demonstrate the effectiveness and superiority of our method, *e.g.*, on NaturalQuestions with an 8x compression constraint, GMSA achieves approximately 36% higher Exact Match (EM) compared to the original input prompt, while also realizing a 2x end-to-end speedup.

## 2 PROBLEM FORMULATION

Given a retrieval-augmented prompt $X = (X^{\text{ins}}, X^{d_1}, ..., X^{d_k}, ..., X^{d_K}, X^{\text{q}})$, where $X^{ins}$, $\{X^{d_k}\}_{k=1}^K$, and $X^{\text{q}}$ represent the instruction, context, and input question respectively. The prompt has a total token length $L$. The key aspect of the context compression system lies in generating a compressed prompt $\widetilde{X}$ with length $\widetilde{L}$, where the compression rate is defined as $\tau = \frac{L}{\widetilde{L}}$. Let $y$ denote the ground truth answer given the original input $X$, and $\widetilde{y}$ denote the answer generated by the large language model (LLM) when input with the compressed prompt $\widetilde{x}$. We aim for the distributions of $y$ and $\widetilde{y}$ to be similar under high compression rates $\tau$. This can be formulated as:

$$\min_{\widetilde{\boldsymbol{x}},\tau} \text{KL} \left( P\left(\widetilde{y} \mid \widetilde{X}\right), P\left(y \mid X\right) \right).$$ (1)

Due to space limitations, we introduce related work in Appendix A.

## 3 GMSA

In this section, we elaborate on our proposed context compression framework, GMSA, which includes two key components: group merging and layer semantic alignment (LSA). GMSA undergoes a two-stage training process: autoencoder training (see Figure 2) and Knowledge Extraction Fine-tuning (KEFT) (see Figure 3). First, GMSA ensures that the generated soft tokens contain the complete semantic representation of the original text through the autoencoder training process. Then, it applies the knowledge contained in the soft tokens to downstream tasks via KEFT.

### 3.1 GROUP MERGING

**Extraction of Semantic Features.** First, we extract the semantic features of the original text through a $k$-layer language modeling model as the encoder. The encoder is trained using LoRA.

$$H = \texttt{Encoder}_k(X),$$ (2)

where $X$ is the original text and $H$ is the obtained last hidden state.

**Merging.** We divide the obtained $H$ into several groups according to the size of the compression limit, as the group length $L_G$ (*e.g.*, when the compression rate is 4, the group length is also 4). To

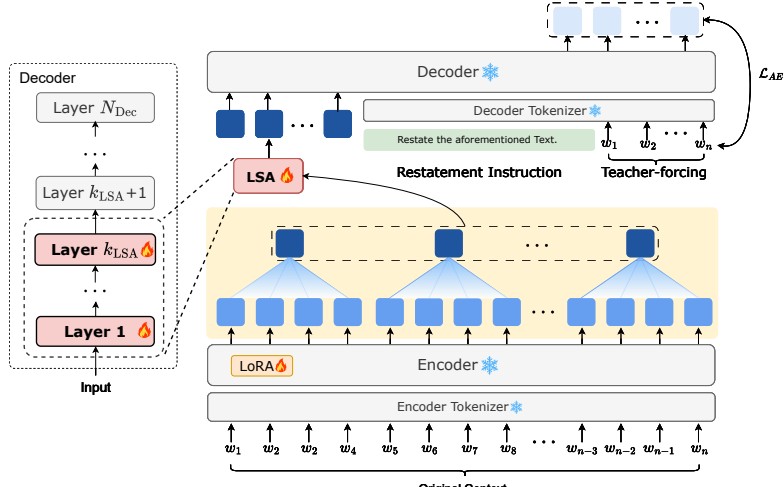

Figure 2: The Autoencoder Training Process of GMSA. GMSA consists of an encoder and a decoder, trained in an autoencoder manner using cross-entropy loss. GMSA first generates a set of summary vectors that meet the compression rate by performing group merging on the last hidden state of the encoder, and then achieves cross-layer semantic alignment through the Layer Semantic Alignment (LSA) module, which is composed of several Transformer blocks initialized with the weights of lower-layer decoder blocks. Remarkably, we find that using just a single layer of LSA can achieve excellent semantic preservation (see Appendix C), hence $k_{\text{LSA}} << N_{\text{Dec}}$.

this end, original text representations are organized as follows:

$$
\begin{aligned}
H &= \left[ H_1, \ldots, H_{\mathbf{G}_j}, \ldots, H_{\mathbf{G}_{N_g}} \right] \\
&= \left[ H_{1:L_G}, \ldots, H_{(j-1) \times L_G : j \times L_G}, \ldots, H_{N_d - L_G + 1 : N_d} \right].
\end{aligned}
$$

We take the average of each dimension of each group token to obtain the initial compressed representation.

$$
\begin{aligned}
\widetilde{H} &= \left[ \bar{H}_{\mathbf{G}_1}, ..., \bar{H}_{\mathbf{G}_i}, ..., \bar{H}_{\mathbf{G}_N} \right] \\
&= \left[ \frac{1}{L_G} \sum H_{\mathbf{G}_1}, ..., \frac{1}{L_G} \sum H_{\mathbf{G}_i}, ..., \frac{1}{L_G} \sum H_{\mathbf{G}_N} \right],
\end{aligned}
$$

where $\widetilde{H}$ is the obtained initial compressed representation.

### 3.2 LAYER SEMANTIC ALIGNMENT

The Layer Semantic Alignment (LSA) module is used to complete the alignment from the soft tokens generated by the encoder (high-level semantics) to the primary semantics of the decoder. Given the significant differences in semantic representation between different layers of LLMs, the LSA is trained via full fine-tuning.

$$
\widetilde{m} = \mathcal{F}_{k_{\text{LSA}}}(\widetilde{H}), \tag{3}
$$

where $H$ is the final compressed representation, $\mathcal{F}_{k_{\text{LSA}}}$ denotes Transformer blocks initialized with the weights from the first $k$ layers of the decoder, and $\widetilde{m}$ denotes the generated soft tokens. Just one layer of LSA is sufficient to achieve excellent semantic preservation (for space limitations, please refer to Appendix C), so in this work, we can just set $k_{\text{LSA}} = 1$.

### 3.3 AUTOENCODER TRAINING

The Autoencoder Training process, which aims to encode the complete information of the original text into memory embeddings, is achieved through autoencoder-based training. We hope to mini-

mize the loss of the reconstructed text, which can be expressed as:

$$\mathcal{L}_{AE} = -\sum_{i=1} \log p_\phi \left( x_i \mid \widetilde{m}, X^{\text{ins}}, x_{<i} \right),$$ (4)

where $p_\phi(\cdot)$ is the `Decoder` probability distribution obtained after the softmax function, and $x_i$ is the $i$-th token in the original text.

### 3.4 KNOWLEDGE EXTRACTION

#### 3.4.1 KNOWLEDGE EXTRACTION PROCESS

Through autoencoder training, we can ensure that the soft tokens obtained via the encoder and LSA contain complete semantic information. Therefore, the next challenge to address is: *how to extract knowledge from the existing soft tokens?*

To guarantee that the generated soft tokens always retain adequate information, we **freeze** the encoder and LSA during the knowledge extraction process, allowing the decoder to complete Knowledge Extraction (KE). Due to space limitations, we elaborate on the differences between KEFT and the recent work LLoCO (Tan et al., 2024) in Appendix I.

We only train the decoder's self-attention module. As shown in Figure 3, the $i$-th token decoding progress can be formulated as:

$$\texttt{Decoder}(\underbrace{\tilde{m}_1, \tilde{m}_2, \tilde{m}_3, \tilde{m}_4, ..., \tilde{m}_{k-1}, \tilde{m}_k}_{\text{soft tokens from the encoder}}, \underbrace{q_1, q_2, ..., q_n}_{\text{question tokens}}, \underbrace{a_1, a_2, ..., a_{i-1}}_{\text{answer tokens}}).$$ (5)

Let $d$ denote the decoder's hidden size, $H \in \mathbb{R}^{(k+n+i-1)\times d}$ denote input hidden states to the self-attention module of the decoder in an arbitrary layer. The above hidden states will be projected into queries, keys, and values as follows:

$$\boldsymbol{Q} = \boldsymbol{W}_Q H, \quad \boldsymbol{K} = \boldsymbol{W}_K H, \quad \boldsymbol{V} = \boldsymbol{W}_V H,$$ (6)

where $\boldsymbol{W}_Q$, $\boldsymbol{W}_K$, and $\boldsymbol{W}_V$ are the projection heads for knowledge extraction. Thus, we now formally present our self-attention computation:

$$\boldsymbol{V'} = \text{softmax}\left(\text{mask}\left(\frac{\boldsymbol{Q}\boldsymbol{K}^T}{\sqrt{d}}\right)\right)\boldsymbol{V},$$ (7)

where $\boldsymbol{V'}$ denotes the output of the self-attention mechanism, which is a refined, context-aware representation of the input values $\boldsymbol{V}$ after applying attention weights.

#### 3.4.2 KNOWLEDGE EXTRACTION FINE-TUNING

After completing autoencoder training, we need to teach the decoder how to utilize the soft tokens. We achieve this by performing full fine-tuning of the $\boldsymbol{W_Q}$, $\boldsymbol{W_K}$, and $\boldsymbol{W_V}$ projection matrices in each layer of the decoder, which can be specifically expressed as:

$$\mathcal{L}_{\text{KE}} = -\sum_{i=1}^{n} \log p_\phi \left( a_i \mid \widetilde{m}, q_1, q_2, ..., q_n, a_{<i} \right),$$ (8)

where $p_\phi(\cdot)$ is the decoder probability distribution obtained after the softmax function, and $a_i$ denotes the $i$-th token in the predicted answer.

## 4 EXPERIMENTS

In this section, we attempt to answer the following research questions (RQs): (1) How effective is GMSA in context restoration? (RQ1) (2) How does GMSA utilize knowledge compared with other baselines? (RQ2) (3) How effective are the individual components of GMSA? (RQ3)

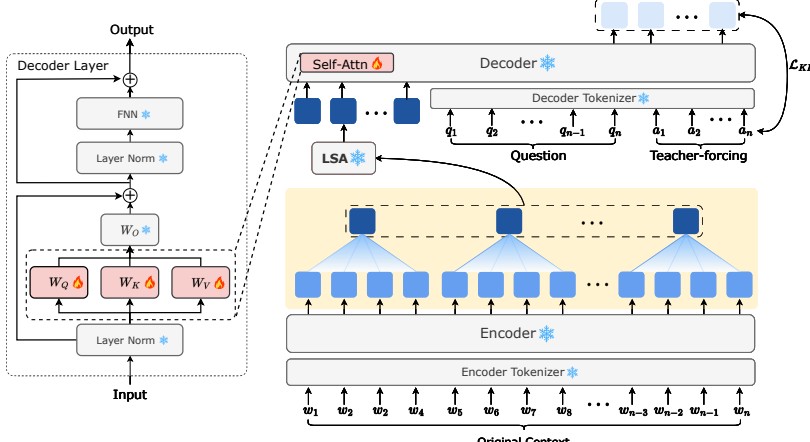

Figure 3: The process of Knowledge Extraction Fine-tuning (KEFT). By fine-tuning only the $W_Q$, $W_K$, and $W_V$ in the attention module of the decoder while keeping other modules frozen, the decoder performs teacher-forcing training using soft tokens $\tilde{m}$, question tokens, and the ground truth answer.

## 4.1 SETTINGS

**Training.** GMSA involves a two-stage training process: autoencoder training and knowledge extraction fine-tuning (KEFT). We utilize seven datasets: PwC (Ge et al., 2024), NaturalQuestions (Liu et al., 2024a), 2WikiMQA (Ho et al., 2020), HotpotQA (Yang et al., 2018), MMLU (Hendrycks et al., 2021a;b), NarrativeQA (Kočiský et al., 2018), GSM8K (Cobbe et al., 2021), and CNN/DailyMail (See et al., 2017) (for more details about the datasets, please refer to Appendix E). Among these, we use PwC to evaluate the performance of context restoration, while the other datasets are used to measure knowledge application. The experimental results in Table 1 are obtained by training GMSA on a mixed dataset composed of NaturalQuestions, 2WikiMQA, and HotpotQA, whereas results on other datasets are obtained by training GMSA on their respective training datasets. During training, we randomly sample compression rates (i.e., 4x compression and 8x compression) for each training sample. Due to space constraints, detailed training settings can be found in Appendix D.

**Implementation.** Built on LLaMA-2-7B (Chat) and Qwen2-7B (Instruct). The maximum lengths for various trainings can be found in Appendix D. All baselines re-implemented from official code for fair comparison.

**Evaluation Metrics.** For the context restoration task on the PwC dataset, we employ BLEU (Papineni et al., 2002), Prefix Exact Match, BERT Score (Zhang* et al., 2020), and ROUGE (Lin, 2004). For the QA tasks across NaturalQuestions, HotpotQA, and 2WikiMQA, we utilize Accuracy (Acc) (Liu et al., 2024a), Exact Match (EM) (Lewis et al., 2020), and F1 score (Yang et al., 2018). We adopt the repository-provided metrics for MMLU (Hendrycks et al., 2021a;b). For other datasets, we use BERT Score for CNN/DailyMail, Accuracy (Acc) for GSM8K, and F1 score for NarrativeQA.

**Baselines.** For the task of context restoration, we train a **T**raditional **C**ompression **P**aradigm **A**uto**E**ncoder (i.e., TCP-AE, see Appendix F for details) as a baseline, employing autoencoder training and the same training hyperparameters as GMSA. We conduct comprehensive comparisons with various methods in text compression and KV-cache compression fields on NaturalQuestions, 2WikiMQA, and HotpotQA, including: hard prompt compression (e.g., LongLLMLingua (Jiang et al., 2024), LLMLingua-2-large (Pan et al., 2024)), soft prompt compression (e.g., AutoCompressor (Chevalier et al., 2023), ICAE (Ge et al., 2024)), and KV-cache compression approaches (e.g., StreamLLM (Xiao et al., 2023), SnapKV (Li et al., 2024), Activation Beacon (Zhang et al., 2024)).

We also further compare with the strong baseline Activation Beacon on CNN/DailyMail, MMLU, GSM8K, and NarrativeQA.

Table 1: Experimental results on three QA benchmark datasets. We **bold** the optimal and underline the suboptimal of baselines. **Acc** refers to accuracy, **EM** refers to exact match, and **F1** denotes the F1 score. **Closed-book** indicates using only the input question as the input, while **Original Prompt** indicates using all retrieved documents as the input. All backbones in this experiment are LLaMA-2-7B, as some important baselines (*e.g.*, Autocompressor, ICAE, and StreamLLM) are only available in LLaMA-2-7B implementations.

| Methods | NaturalQuestions | | | 2WikiMQA | | | HotpotQA | | |
| --- | --- | --- | --- | --- | --- | --- | --- | --- | --- |
| | Acc | EM | F1 | Acc | EM | F1 | Acc | EM | F1 |
| Closed-book | 24.14 | 20.23 | 21.88 | 25.37 | 24.96 | 27.82 | 18.34 | 17.22 | 24.02 |
| Original Prompt | 55.40 | 15.07 | 26.81 | 37.54 | 30.84 | 37.79 | 44.21 | 34.35 | 47.49 |
| *4x compression constraint* | | | | | | | | | |
| *KV-cache Compression Methods* | | | | | | | | | |
| StreamLLM | 29.53 | 7.87 | 15.38 | 28.47 | 26.49 | 30.78 | 28.90 | 23.87 | 34.32 |
| SnapKV | 58.64 | 12.58 | 23.07 | 29.86 | 27.61 | 32.62 | 37.35 | 30.51 | 42.08 |
| Activation Baecon | 56.20 | 25.65 | 34.17 | 34.45 | 24.42 | 32.05 | 44.45 | 25.80 | 39.82 |
| *Prompt Compression Methods* | | | | | | | | | |
| AutoCompressor | 13.79 | 0.00 | 1.34 | 41.56 | 0.00 | 8.07 | 20.98 | 0.01 | 6.80 |
| ICAE | 17.33 | 1.24 | 7.05 | 35.17 | 10.25 | 22.04 | 34.16 | 13.02 | 26.69 |
| LongLLMLingua | 53.41 | 39.62 | 43.03 | 33.88 | 31.71 | 37.05 | 40.31 | 35.55 | 48.68 |
| LLMLingua-2-large | 41.77 | 29.49 | 34.79 | 31.07 | 28.88 | 33.37 | 33.15 | 28.80 | 40.89 |
| **GMSA** | **69.98** | **58.12** | **57.59** | **55.95** | **49.55** | **57.17** | **53.52** | **44.60** | **59.31** |
| *8x compression constraint* | | | | | | | | | |
| *KV-cache Compression Methods* | | | | | | | | | |
| StreamLLM | 31.22 | 7.72 | 14.93 | 27.43 | 25.82 | 29.76 | 26.58 | 21.78 | 32.21 |
| SnapKV | 57.21 | 11.86 | 22.49 | 28.19 | 26.56 | 30.97 | 34.54 | 28.10 | 40.16 |
| Activation Baecon | 51.22 | 23.01 | 31.45 | 33.20 | 25.12 | 32.20 | 40.30 | 24.40 | 37.63 |
| *Prompt Compression Methods* | | | | | | | | | |
| AutoCompressor | 17.51 | 0.00 | 1.63 | 41.76 | 0.00 | 8.09 | 22.04 | 0.00 | 6.93 |
| ICAE | 17.74 | 0.72 | 3.23 | 33.56 | 5.74 | 17.19 | 30.40 | 4.42 | 15.80 |
| LongLLMLingua | 46.55 | 36.65 | 40.72 | 31.53 | 29.93 | 34.08 | 34.73 | 31.60 | 43.85 |
| LLMLingua-2-large | 30.73 | 21.92 | 27.61 | 27.45 | 26.57 | 29.64 | 24.14 | 22.11 | 31.69 |
| **GMSA** | **62.34** | **51.00** | **53.09** | **51.33** | **46.67** | **54.22** | **46.52** | **38.39** | **53.77** |

## 4.2 MAIN RESULT

We highlight the findings of GMSA in two aspects: context restoration and downstream knowledge application.

For RQ1, GMSA-AE significantly outperforms the Traditional Compression Paradigm AutoEncoder (TCP-AE) in context restoration. (Due to space constraints, we defer the detailed analysis to Appendix B). In terms of quality (Figure 4), GMSA-AE surpasses TCP-AE by over 20% on token-matching metrics (BLEU, ROUGE, Prefix EM[2]) and by 5% on semantic similarity (BERT Score F1), indicating superior memory for both precise tokens and overall semantics. Furthermore, GMSA-AE demonstrates substantially faster convergence and greater robustness (Figure 5). It con-

---

[2]Prefix Exact Match represents the ratio of the correctly matched prefix length to the total length. For example, in a 512-token sequence, if the first 128 tokens are an exact match but the 129th token is not, the Prefix Exact Match score is calculated as 128/512 = 0.25.

Table 2: Performance comparison between GMSA and Activation Beacon on CNN / DailyMail.

| Dataset | Backbone | Comp. Constraint | Activation Beacon | GMSA |
|---|---|---|---|---|
| CNN / DailyMail | LLaMA-2-7B | 4x | 87.0 | **89.1** |
| | | 8x | 86.5 | **88.8** |

verges in 1000 steps, while TCP-AE fails to converge even after 5000 steps. Crucially, GMSA-AE's performance is robust to reducing encoder layers—a setting where TCP-AE's performance severely degrades. Appendix K and J provide further evidence, including perplexity scores and case studies.

For RQ2, GMSA's knowledge utilization typically shows significantly better performance than other baselines across various compression rate constraints (see Table 1, Table 2 and Table 3). In the KV-cache compression methods, the compressed representation and the target model must be consistent. Although this avoids the problem of cross-layer semantic alignment, it severely limits the flexibility of applying the compressed representation.

Table 3: Performance on MMLU and GSM8K.

| Dataset | Methods | Backbone | |
|---|---|---|---|
| | | LLaMA-2-7B | Qwen2-7B |
| MMLU | Original Prompt | 46.3 | 65.6 |
| | Activation Beacon | 45.1 | 64.3 |
| | **GMSA** | **47.6** | **67.3** |
| GSM8K | Original Prompt | 27.6 | 81.3 |
| | **Activation Beacon** | **27.8** | **81.9** |
| | GMSA | 26.4 | 80.5 |

Compared with the KV-cache compression methods (*i.e.*, streamLLM, SnapKV, and Activation Beacon), GMSA achieves the best performance while maintaining flexibility. In contrast to prompt-based compression algorithms, whether they are query-independent prompt compression algorithms (*i.e.*, ICAE, AutoCompressor, and LLMLingua-2-large) or query-dependent LongLLMLingua, their performance is far below that of GMSA. It is worth noting that GMSA adopts a query-independent compression mechanism and still significantly outperforms the query-dependent LongLLMLingua, which sufficiently illustrates the effectiveness and superiority of GMSA. We further evaluate GMSA on a diverse set of tasks, including summarization (CNN / DailyMail), general knowledge (MMLU) and mathematical reasoning (GSM8K), with the strong baseline Activation Beacon.

As results in Table 2 and Table 3, GMSA demonstrates robust performance across tasks. On generation-centric task such as summarization, GMSA consistently outperforms Activation Beacon under both 4x and 8x compression contraint. We directly evaluate the decoder of GMSA (after two-stage training) on multidisciplinary benchmark (MMLU) and mathematical reasoning (GSM8K) to assess its ability to retain general knowledge in short texts. On MMLU, GMSA not only surpasses the baseline but even outperforms the original uncompressed input, suggesting that compression may help focus on core semantics. A slight performance drop is observed on GSM8K, which we attribute to the lack of mathematical-domain data in GMSA's training corpus.

Even in ultra-long scenarios (NarrativeQA, see Appendix H.2), GMSA not only achieves a 2x speedup over the original prompt input, but also attains substantially higher F1 scores.

## 4.3 EFFICIENCY ANALYSIS

In this section, we discuss the efficiency of our proposed method. By using soft tokens instead of the long original context to enhance the inference process, our method reduces the inference cost of the original context during the generation process by a factor of $r$. The overall floating-point operations (FLOPs) are calculated through two processes: compression and generation.

The compression process can be expressed as:

$$\text{FLOPs}^{comp} = F^{\text{Encoder}}(L) + F^{\text{LSA}}\left(\left\lceil \frac{L}{r} \right\rceil\right)$$

Here, $L$ denotes the original context length, $L_q$ denotes the question length, and $F^*(\cdot)$ represents the FLOPs complexity measure for module $*$, with the specific calculation process detailed in Appendix G. The symbol $*$ indicates the architectural components, where $* \in \{\text{Decoder}, \text{Encoder}, \text{LSA}\}$. For the generation process, assuming the answer length is $L_a$, the generation process requires $L_a$ forward passes. The FLOPs for the $i$-th forward pass are given by:

$$\text{FLOPs}_i^{forward} = F^{\text{Decoder}}\left(\left\lceil \frac{L}{r} \right\rceil, L_q, i\right)$$

Combining the costs of all components, the total FLOPs complexity is:

$$\text{FLOPs} = \sum_{i=1}^{L_a} \text{FLOPs}_i^{\text{forward}} + \text{FLOPs}^{comp}$$

Thanks to the ability to retain complete semantics with only a few encoder layers (*e.g.*, 8 layers), GMSA achieves the lowest end-to-end inference latency, which is approximately 2x faster than other methods on NaturalQuestions and NarrativeQA (see Appendix H).

## 4.4 ABLATION STUDY

For RQ3, to investigate the impact of each component in GMSA, we conduct the following four ablation experiments (see Table 4): (1) Ours w/o Autoencoder Training refers to performing knowledge extraction fine-tuning on GMSA directly without knowledge memory training. (2) Ours w/o Knowledge Extraction Fine-tuning means only performing Autoencoder-Training on GMSA. (3) Ours w/o Group Merging indicates that we replace group merging with appending meaningless learnable tokens when generating summary vectors. (4) Ours w/o Layer Semantic Alignment means we do not use the Layer Semantic

Table 4: The impact of different components in GMSA on the PwC test set under 4x compression constraint, measured by BERT Score F1.

| Method | BERT Score F1 |
|---|---|
| **Default** | **0.91** |
| w/o Autoencoder Training | 0.87 |
| w/o Knowledge Extraction Fine-tuning | 0.83 |
| w/o Group Merging | 0.82 |
| w/o Layer Secmantic Alignment | 0.84 |
| w Qwen2-7B-Instruct | 0.90 |

Alignment module and directly employ summary vectors as soft tokens. (5) Ours w/ Qwen2-7B-Instruct refers to replacing the decoder with Qwen2-7B-Instruct.

In summary, the removal of any single component leads to a significant drop in performance, which fully demonstrates the necessity and effectiveness of each component. Removing Autoencoder Training makes it difficult for GMSA to generate summary vectors that encompass complete semantics, while eliminating Knowledge Extraction Fine-tuning causes GMSA to lose its ability to extract knowledge in downstream tasks, both of which would deteriorate performance. Replacing Group Merging with appending learnable tokens would increase the difficulty of learning, and discarding the LSA module would result in misalignment between the high-level semantic information represented by summary vectors and the low-level semantic space of the decoder's input. When the encoder and decoder are different, GMSA can still maintain high performance, which fully demonstrates its robustness and generalization ability.

## 5 CONCLUSION

This paper introduces GMSA, a context compression framework based on an encoder-decoder structure. It evenly and efficiently learns summary vectors and bridges the significant gap between the semantics representation of different layers via group merging, and a LSA module. GMSA first undergoes autoencoder training to ensure that the generated soft tokens contain nearly complete semantics, and then adapts to downstream tasks via KEFT. Experiments demonstrate that GMSA converges quickly, can stably converge even with random sampling compression rates for each sample and using only a few encoder layers, and has excellent context restoration capabilities. It outperforms existing baselines by a large margin in downstream tasks, paving the way for the efficient application of LLMs.

ETHICS STATEMENT

This paper introduces GMSA, a context compression framework based on the encoder-decoder architecture. It effectively and efficiently learns summary vectors and bridges the significant gap between different layers via group merging, and a LSA module. The data and models used in our research are released under open-source licenses and sourced from open platforms. Although our work may have various societal impacts, it does not introduce any additional ethical concerns compared to existing text compression methods. Therefore, we believe it is unnecessary to specifically highlight any particular ethical issues here.

REPRODUCIBILITY STATEMENT

Core code implementing GMSA and the baselines is provided in the supplementary material.

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

## A  RELATED WORK

**Hard Prompt Compression.**    Hard prompt compression refers to the removal of some less important tokens from the original prompt or the generation of summaries to achieve compression. The compressed prompt is explicit text. It can mainly be divided into the following four categories: (1) Perplexity-based methods. Selective-Context (Li et al., 2023) removes certain lexical units based on perplexity, while methods such as LLMLingua (Jiang et al., 2023), LongLLMLingua (Jiang et al., 2024), and Perception Compressor (Tang et al., 2025) adopt a coarse-to-fine framework to gradually eliminate less important parts. (2) Bidirectional semantic-based methods. Considering the unidirectional nature of perplexity, some approaches employ bidirectional semantic information for compression, such as LLMLingua-2 (Pan et al., 2024), MOOSComp (Zhou et al., 2025), and EFPC (Cao et al., 2025). (3) Methods based on intrinsic attention mechanisms. Compression is achieved through the intrinsic attention mechanisms of LLMs, such as PIS (Chen et al., 2025) and AttnComp (Zhao et al., 2025). (4) Summary generation. This involves generating linguistic summaries that contain useful information for long text content, such as CompACT (Yoon et al., 2024) and RECOMP (Xu et al., 2024). *Although these methods improve the computational efficiency of inference through prompt compression, they compromise the semantic integrity of the original prompt.*

**Soft Prompt Compression.**    Soft prompt compression has become a research hotspot in the field of Natural Language Processing (NLP). The goal of soft prompt compression is to learn a set of soft tokens (with a sequence length much shorter than the original text) to achieve compression, where the compressed soft prompts cannot be explicitly converted into text. Among them, xRAG (Cheng et al., 2024) focuses on processing short texts and extreme compression. More recent methods learn soft tokens by appending randomly initialized learnable tokens, including GIST (Mu et al., 2023), AutoCompressor (Chevalier et al., 2023), 500xCompressor (Li et al., 2025), ICAE (Ge et al., 2024), LLoCO (Tan et al., 2024), and others (Ye et al., 2024; Liao et al., 2025; Dai et al., 2025; Rau et al., 2025; Choi et al., 2025). This leads to the semantics of anchor tokens in the input sequence being increasingly emphasized layer by layer, while the semantics of other tokens are diluted and cannot be fully preserved in the summary vectors. Moreover, these methods only use Multi-Layer Perceptrons (MLPs) for coarse-grained semantic alignment when semantic alignment is required, ignoring the significant differences in representations across different layers of large models. *Our proposed method evenly and effectively extracts summary vectors through group merging. By employing a group average pooling merging strategy, it addresses the issue of uneven semantic learning. Additionally, it bridges the large semantic gap between different layers of LLMs through a Layer Semantic Alignment (LSA) module.*

**KV-cache Compression.**    Research in this direction focuses on directly compressing the KV-cache in each transformer layer, considering factors such as layer-wise compression, attention heads, the importance of different KVs, or token-level approaches. Examples include CLA (Brandon et al., 2024), which shares KV-cache across layers; GQA (Ainslie et al., 2023) and MQA (Shazeer, 2019), which reduce the number of heads for keys and values; StreamLLM (Xiao et al., 2023) and SnapKV (Li et al., 2024), which discard unimportant KVs for efficient compression; and Activation Beacon, which appends some meaningless tokens (shorter than the original length) and learns compressed representations in the KV-cache of these tokens for each layer. While KV-cache-based compression methods can accelerate inference, they require the compression and response models to be identical. *This limitation restricts practical applications and increases resource consumption, e.g., in prompt compression for large models (e.g., 70B), a smaller model (e.g., 7B) cannot be used as the compression model; instead, the same oversized model must be employed.*

## B  CONTEXT RESTORATION CAPABILITY

In the context restoration task, GMSA-AE significantly outperforms the **T**raditional **C**ompression **P**aradigm **A**uto**E**ncoder (TCP-AE) in multiple aspects, including restoration quality (see Figure 4), convergence speed, and robustness (see Figure 5). As shown in Figure 4, GMSA-AE outperforms TCP-AE in all evaluation metrics. BLEU Score (Papineni et al., 2002), Prefix Exact Match, and ROUGE (Lin, 2004) are token-matching-based metrics, and GMSA-AE's performance in these metrics is at least 20% higher than TCP-AE under all compression constraints, indicating that GMSA-AE has a stronger ability to precisely remember each token. The BERT Score F1 (Zhang* et al.,

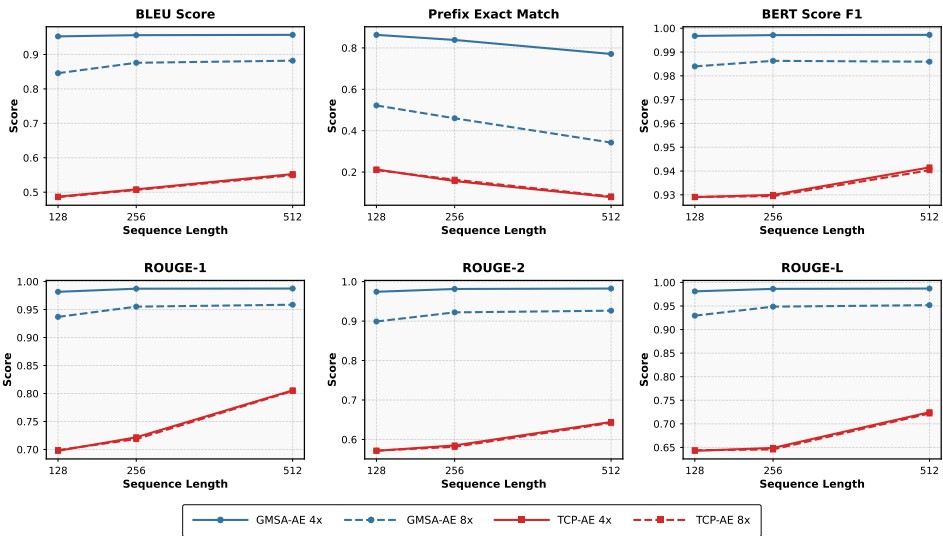

Figure 4: GMSA-AE v.s. TCP-AE on the context restoration task. Sequence Length represents different context restoration lengths (*i.e.*, 128, 256, 512), and the models are trained with a maximum length of 512.

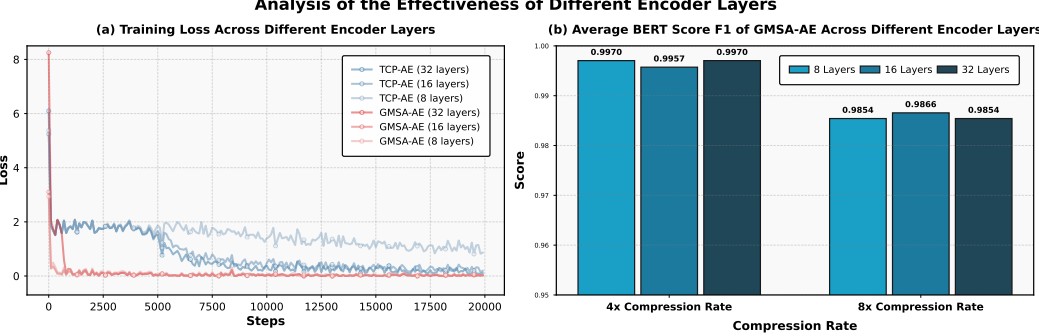

Figure 5: Analysis of the Effectiveness of Different Encoder Layers. (a) represents the comparison of convergence speed between GMSA-AE with different encoder layers and TCP-AE. (b) denotes the impact of different encoder layers on the semantic retention of GMSA-AE. The average BERT Score F1 refers to the average F1 score across different context restoration lengths (*i.e.*, 128, 256, and 512).

2020), which measures semantic similarity and reflects the model's ability to remember overall semantics, is also about 5% higher for GMSA-AE than TCP-AE. As shown in Figure 5, GMSA-AE converges much faster than TCP-AE. GMSA-AE convergence around 1000 training steps, while TCP-AE has not fully converged even after 5000 steps. Moreover, significantly reducing the number of encoder layers (*e.g.*, to 8 encoder layers) makes TCP-AE converge much more slowly. In contrast, GMSA-AE demonstrates robustness under different settings. In terms of convergence speed, reducing the number of encoder layers even further accelerates the convergence of GMSA-AE: versions with 8 or 16 encoder layers converge faster than those with 32 layers, possibly because the cross-layer semantic alignment challenge is alleviated with fewer encoder layers. From the perspective of semantic retention, the Average BERT Score F1 of different encoder layers remains consistent under various compression rates, indicating that even with a small number of encoder layers (*e.g.*, 8 layers), GMSA-AE can still effectively retain semantic information and complete high-quality memory tasks. We also evaluate the quality of the reconstructed text using perplexity, and the results show that GMSA-AE significantly outperforms TCP-AE (see Appendix K). Moreover, we conduct

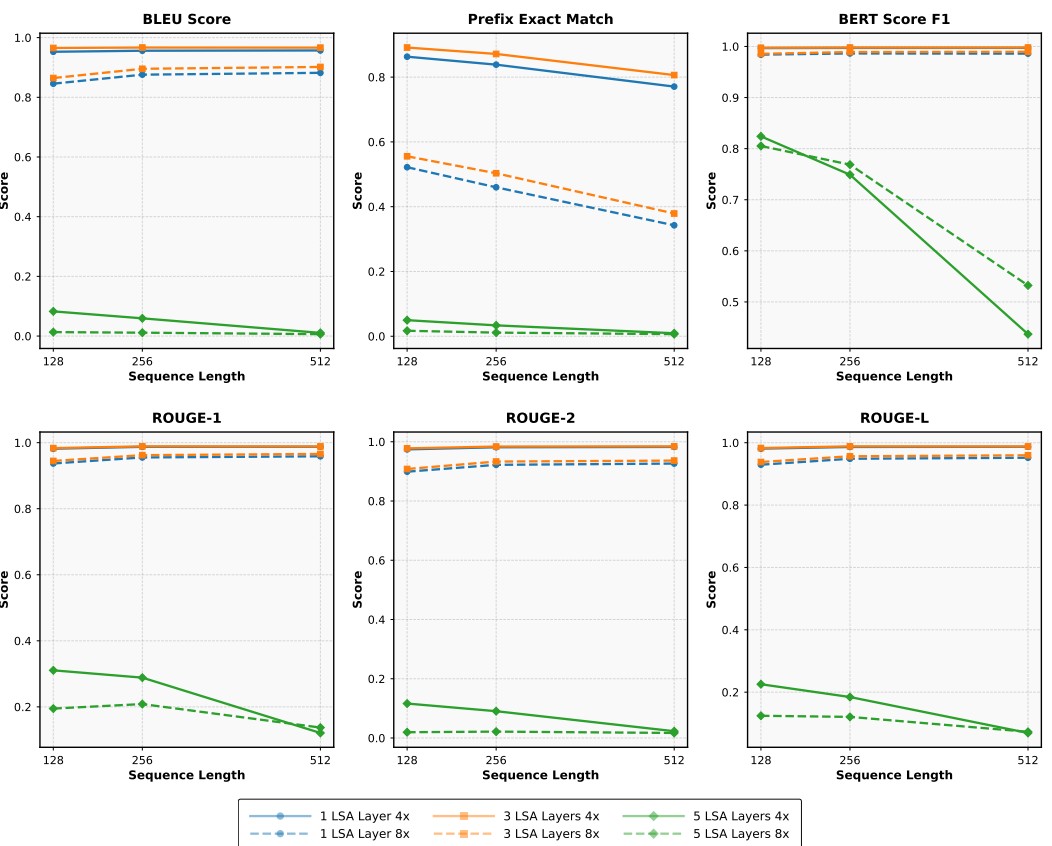

Figure 6: The impact of different layers of LSA on semantic retention in GMSA-AE. Sequence Length represents different context restoration lengths (*i.e.*, 128, 256, 512), and the model is trained with a maximum length of 512.

specific case studies to further verify the performance gap between GMSA-AE and TCP-AE (see Appendix J).

## C  IMPACT OF DIFFERENT LAYER SEMANTIC ALIGNMENT LAYERS

We conduct experiments to investigate the impact of layer semantic alignment (LSA) module with varying numbers of layers on the retention of complete semantics, and the results are shown in Figure 6. We can draw the following conclusions: (1) Only one layer of LSA is sufficient to achieve good retention of complete semantics (with a BERT Score F1 close to 1, and it already performs the best among different numbers of LSA layers). (2) When the number of LSA layers becomes too high, *e.g.*, using five layers of LSA, it may actually lead to a decrease in the GMSA's ability to retain semantics. This is likely because as the LSA module becomes deeper, it contains more high-layer semantics and fewer low-layer semantics, thereby increasing the difficulty of semantic alignment.

## D  IMPLEMENTATION DETAILS

We train GMSA on two NVIDIA A100 GPUs (80GB) using bf16 precision. For the PwC dataset, we train on the full dataset with 10,000 steps for Autoencoder Training and 5,000 steps for Knowledge Extraction Fine-tuning (KEFT). For the QA datasets (*i.e.*, NaturalQuestions, 2WikiMQA, and HotpotQA), we sample 15,000 examples from each to form the training set, using 20,000 steps for Autoencoder Training and 1,000 steps for KEFT, respectively. Other parameters are listed in Table 5.

Table 5: Training Hyperparameters.

| Hyperparameter | Value |
|---|---|
| Optimizer | AdamW |
| Learning Rate | $1 \times 10^{-4}$ (Autoencoder Training) $1 \times 10^{-5}$ (KEFT) |
| Batch Size | 4 (Autoencoder Training) 16 (KEFT) |
| Scheduler | Linear |
| Gradient Clip Norm | 2.0 |

The relationship between the maximum training token length and the dataset is shown in Table 6.

Table 6: The relationship between the maximum token length and the dataset.

| Dataset | Maximum Training Token Length |
|---|---|
| PwC | 512 |
| CNN / Daily | 1024 |
| NaturalQuestions, 2WikiMQA, HotpotQA | 3072 |
| NarrativeQA | 32768 |

## E DATASETS DETAILS

**PwC dataset.** In the PwC dataset (Ge et al., 2024), each sample is a triplet (context, prompt, answer), where the context is sampled from the Pile and the prompt and answer are generated by GPT-4. The training set contains 241,564 samples, the test set contains 18,146 samples, and the average token length of the dataset is 609[3].

**NaturalQuestions.** NaturalQuestions (Liu et al., 2024a), in which each question corresponds to 20 relevant documents, 19 of which are distractors and only one contains the ground truth answer. The training set contains 75,322 samples, the test set contains 2,655 samples, and the average token length of the dataset is 3,253.

**HotpotQA.** HotpotQA (Yang et al., 2018) is a two-hop reasoning dataset, where the answers are scattered across two documents. Specifically, each question corresponds to 10 relevant documents, two of which are the ground truth documents. The training set contains 89,609 samples, the test set contains 7,345 samples, and the average token length of the dataset is 1,567.

**2WikiMQA.** Compared with HotpotQA, 2WikiMQA (Ho et al., 2020) includes more complex reasoning paths, and the combination of structured and unstructured data, usually involving two or more hops and having higher difficulty. The training set contains 167,454 samples, the test set contains 12,576 samples, and the average token length of the dataset is 1098.

**MMLU.** MMLU (Hendrycks et al., 2021a;b) is a benchmark designed to measure a language model's knowledge and problem-solving abilities across a wide range of subjects, including humanities, social sciences, STEM, and more. It consists of multiple-choice questions that cover 57 diverse tasks, aiming to evaluate a model's general competence in understanding and responding to complex prompts that require factual recall, reasoning, and common sense.

**GSM8K.** GSM8K (Cobbe et al., 2021) is a dataset of elementary mathematics word problems, specifically curated to test the reasoning capabilities of language models. Each problem requires multi-step arithmetic operations and logical deduction to arrive at the correct answer. The dataset is

---

[3]We uniformly use the tokenizer of LLaMA-2-7B (chat) to calculate the token length of the text.

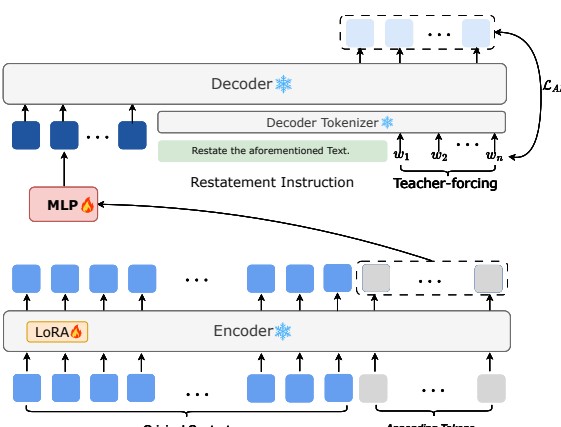

Figure 7: The training process of Traditional Compression Paradigm Autoencoder (TCP-AE). The traditional compression paradigm first adds appending tokens after the Original Context, then employs an encoder (*e.g.*, LLaMA) to autoregressively learn summary vectors. These summary vectors are then processed through a Multilayer Perceptron (MLP) layer to achieve coarse-grained semantic alignment, resulting in soft tokens. On the decoder side, context restoration training is conditioned on soft tokens, with cross-entropy used as the final loss.

designed to be challenging enough to distinguish between models with strong quantitative reasoning skills and those that struggle with sustained logical deduction.

**NarrativeQA.** NarrativeQA (Kočiský et al., 2018) is a question-answering dataset focused on understanding long-form narratives. It comprises pairs of books and questions about their content, where the answers often require synthesizing information from multiple parts of the text. The dataset aims to assess a model's ability to comprehend complex storylines, identify key characters and events, and answer questions that go beyond simple fact retrieval.

**CNN / DailyMail.** CNN / DailyMail (See et al., 2017) is a popular dataset for abstractive summarization, consisting of news articles from CNN and the Daily Mail. Each article is paired with a human-written summary, acting as the ground truth. The task involves generating a concise and coherent summary of the input news article, requiring models to identify the most important information and rephrase it effectively.

## F   TRADITIONAL COMPRESSION PARADIGM AUTOENCODER TRAINING

As shown in Figure 7, to fully measure the context restoration capability of GMSA after Autoencoder Training, we conduct Autoencoder Training following the traditional compression paradigm, using the same training method as GMSA (*i.e.*, randomly sampling compression rates for training examples and other hyperparameters in the training process are also the same) to obtain Traditional Compression Paradigm Autoencoder (TCP-AE)[4].

## G   FLOPS CALCULATION

Let $L_{\text{in}}$ denote the input sequence length. We calculate the floating-point operations (FLOPs) for a single layer can be decomposed into Attention and Feed Forward Network (FFN) operations. The calculation process for the Attention operation is:

---

[4]The entire structure is similar to the pretraining structure of ICAE, but the training paradigm is different. For example, we randomly sample the compression rate for training, which increases the difficulty of training.

$$F^{Attention}(L_{\text{in}}) = F^{qkv}(L_{\text{in}}) + F^{qk}(L_{\text{in}}) + F^{softmax}(L_{\text{in}}) + F^{av}(L_{\text{in}}) + F^{out}(L_{\text{in}}),$$

$$F^{qkv}(L_{\text{in}}) = 2 \times L_{\text{in}} \times D \times d \times h^q + 2 \times 2 \times L_{\text{in}} \times D \times d \times h^k,$$

$$F^{qk}(L_{\text{in}}) = 2 \times h^q \times L_{\text{in}} \times L_{\text{in}} \times d,$$

$$F^{softmax}(L_{\text{in}}) = h^q \times L_{\text{in}} \times L_{\text{in}},$$

$$F^{av}(L_{\text{in}}) = 2 \times h^q \times L_{\text{in}} \times L_{\text{in}} \times d,$$

$$F^{out}(L_{\text{in}}) = 2 \times L_{\text{in}} \times d \times h^q \times D. \tag{9}$$

The calculation process for the FFN can be formulated as:

$$F^{FFN}(L_{\text{in}}) = F^{up}(L_{\text{in}}) + F^{down}(L_{\text{in}}),$$

$$F^{up}(L_{\text{in}}) = 2 \times L_{\text{in}} \times D \times 2 \times I,$$

$$F^{down}(L_{\text{in}}) = 2 \times L_{\text{in}} \times D \times I. \tag{10}$$

Denote the original context length as $L$, the compression rate as $r$, question length as $L_q$, answer length as $L_a$, the number of layers in the LSA as $N_{\text{LSA}}$, the number of decoder layers as $N_{\text{Dec}}$, the number of encoder layers as $N_{\text{Enc}}$, query head number as $h^q$, key/value head number as $h^k$, the hidden size as $D$, head dimension as $d$, intermediate size as $I$, and vocabulary size as $V$. Therefore, the FLOPs of the encoder, LSA, and decoder can be expressed as:

$$F^{Encoder}(L) = \left( F^{Attention}(L) + F_E^{FFN}(L) \right) \times N_{\text{Enc}},$$

$$F^{LSA}(\lceil L/r \rceil) = \left( F_L^{Attention}(\lceil L/r \rceil) + F_L^{FFN}(\lceil L/r \rceil) \right) \times N_{\text{LSA}},$$

$$F^{Decoder}\left(\lceil L/r \rceil, L_q, L_a\right) = \sum_{i=1}^{L_a} \left( F_D^{Attention}\left(\lceil L/r \rceil, L_q, i\right) + F_D^{FFN}\left(\lceil L/r \rceil, L_q, i\right) \right) \times N_{\text{Dec}}. \tag{11}$$

where $N_{\text{Enc}} \ll N_{\text{total}}$ uses only shallow layers (*e.g.*, 8/32 in LLaMA), $N_{\text{LSA}}$ is generally set to 1 follows from LSA's layer-agnostic property (see Appendix C), and $r > 1$ represents standard compression rates.

# H    LATENCY EVALUATION

## H.1    EFFICIENCY ANALYSIS ON GENERAL SCENARIOS

We conduct an empirical test on the NaturalQuestions to evaluate the impact of GMSA on inference efficiency under 4x and 8x compression constraints.[5] In this efficiency test, we fix the generation length to 100. Table 7 shows that the context compression by GMSA helps improve the inference efficiency of LLMs. Compared with all settings, including the original prompt, Kv-cache compression algorithms (*i.e.*, StreamLLM, SnapKV, and Activation Beacon), and the encoder-decoder-based ICAE, GMSA achieves more than a 2x end-to-end inference speedup.

## H.2    EFFICIENCY ANALYSIS ON ULTRA-LONG SCENARIOS

To evaluate GMSA on ultra-long scenarios, we conduct experiments on NarrativeQA (Kočiský et al., 2018) with

Table 8: Performance and Latency on NarrativeQA (32K max length, Qwen2-7B as backbone).

| Method | F1 | Latency (s) |
|---|---|---|
| Original Prompt | 9.7 | 5.2 |
| *4x compression constraint* | | |
| **GMSA** | **15.5** | **2.7** |
| *8x compression constraint* | | |
| **GMSA** | **14.1** | **2.3** |

---

[5]We test the latency on two NVIDIA A800 GPUs (80G).

Table 7: Latency Evaluation. Latency evaluation of different methods under varying compression constraints on the Natural Questions dataset. The symbol ✗ indicates that the specific processing time is unavailable.

| Methods | Compression Time | Decoding Time | End-to-End Inference Time |
|---|---|---|---|
| Original Context | - | 1.14 | 1.14 |
| *4x compression constraint* | | | |
| StreamLLM | ✗ | ✗ | 1.47 |
| SnapKV | ✗ | ✗ | 0.99 |
| Activation Beacon | ✗ | ✗ | 3.06 |
| ICAE | 0.73 | 1.06 | 1.79 |
| **GMSA** | **0.27** | **0.18** | **0.45** |
| *8x compression constraint* | | | |
| StreamLLM | ✗ | ✗ | 1.41 |
| SnapKV | ✗ | ✗ | 0.99 |
| Activation Beacon | ✗ | ✗ | 1.92 |
| ICAE | 0.56 | 2.60 | 3.16 |
| **GMSA** | **0.27** | **0.15** | **0.42** |

a maximum context length of 32K tokens, using Qwen2-7B as the backbone[6]. We report end-to-end inference latency (in seconds).

As shown in Table 8, GMSA achieves significant acceleration. Under both 4× and 8× compression rates, GMSA is 2x faster than processing the original prompt while attaining substantially higher F1 scores.

This demonstrates that, despite the quadratic complexity retained in a few layers of the encoder, the overall end-to-end efficiency gain from compressing the context into a small set of soft tokens is substantial—even for ultra-long sequences.

# I  COMPARISON WITH LLoCO

We provide a detailed comparison with the recent work LLoCO (Tan et al., 2024), which also employs an encoder-decoder architecture and a decoder-only fine-tuning strategy for downstream tasks, conceptually similar to our Knowledge Extraction Fine-tuning (KEFT).

Despite this high-level similarity, GMSA introduces several key structural and methodological innovations that lead to significant performance improvements:

**Group Merging.** GMSA proposes a novel Group Merging strategy to evenly retain semantics from the original context. By dividing the encoder's last hidden state into groups and applying average pooling, this method effectively mitigates the problem of uneven semantic learning, where the semantics of anchor tokens are disproportionately emphasized at the expense of others. This is a common limitation in autoregressive summary vector learning approaches, including LLoCO.

Table 9: Performance comparison with LLoCO on the NaturalQuestions (LLaMA-2-7B as backbone).

| Method | Acc | EM | F1 |
|---|---|---|---|
| LLoCO | 41.7 | 38.1 | 39.1 |
| *4x compression constraint* | | | |
| **GMSA** | **70.0** | **58.1** | **57.6** |
| *8x compression constraint* | | | |
| **GMSA** | **62.3** | **51.0** | **53.1** |

---

[6]To avoid out-of-memory issues on ultra-long scenarios, we evaluate latency on two NVIDIA H20 GPUs (94GB).

**Layer Semantic Alignment (LSA).** A core innovation of GMSA is the explicit Layer Semantic Alignment module. This component is designed to bridge the large semantic gap between the high-level, abstract summary vectors generated by the encoder and the low-level semantic space expected by the decoder's input layers. LLoCO does not incorporate such a dedicated mechanism for cross-layer semantic alignment.

**Knowledge Extraction Fine-tuning (KEFT).** The KEFT process in GMSA is specifically designed to fine-tune *only* the weight matrices $W_Q$, $W_K$, and $W_V$ in the self-attention modules of each decoder layer. This design is motivated by the understanding that the attention mechanism primarily governs information flow and context integration, while the feed-forward network (FFN) acts more as a static knowledge storage module (Geva et al., 2021). By selectively tuning only the attention projections, KEFT efficiently adapts the decoder to extract task-specific knowledge from the compressed soft tokens without altering the core knowledge representations. In contrast, LLoCO applies Low-Rank Adaptation (LoRA) to the entire decoder, including both attention and FFN components.

To provide a quantitative comparison, we trained and evaluated LLoCO on the NaturalQuestions (NQ) dataset using its official open-source code and default settings, with LLaMA-2-7B as the backbone model. The results, presented in Table 9, demonstrate the superior performance of GMSA.

This comparison clearly highlights the effectiveness of GMSA's architectural components in achieving state-of-the-art results for context compression and knowledge extraction.

## J Perplexity Evaluation

For the task of context restoration, we evaluate model performance from the perspective of perplexity. The experimental results are shown in Table 10. "Condition Type" represents the basic conditions under which the LLMs recovers the text, which are divided into three types: recovering from the Original Context, recovering from the soft tokens generated by TCP-AE, and recovering from the soft tokens generated by GMSA-AE. Different Sequence Lengths represent different lengths of the context restoration task. We can draw two key findings: (1) Under different compression constraints and restoration lengths, the perplexity of the recovered text conditioned on TCP-AE-generated soft tokens is significantly higher than that of the recovered text conditioned on the Original Context. (2) Except for the case where the compression constraint is 8x and the restoration length is 512, where GMSA-AE's recovered text perplexity is slightly lower than that of the Original Context (by only 0.02), in all other cases, GMSA-AE's recovered text perplexity is lower than that of the Original Context. Furthermore, in all scenarios, GMSA-AE's recovered text perplexity is significantly lower than that of the recovered text conditioned on TCP-AE-generated soft tokens.

Table 10: Comparison of the average token perplexity under different condition types on the PwC test set.

| Condition Type | Sequence Length | | |
|---|---|---|---|
| | 128 | 256 | 512 |
| Original Context | 1.12 | 1.06 | 1.03 |
| *4x compression constraint* | | | |
| TCP-AE | 1.36 | 1.34 | 1.35 |
| **GMSA-AE** | **1.01** | **1.01** | **1.00** |
| *8x compression constraint* | | | |
| TCP-AE | 1.36 | 1.34 | 1.35 |
| **GMSA-AE** | **1.08** | **1.06** | **1.05** |

## K Case Study

As shown in Table 11, we use the restoration of a specific text to study the performance of GMSA-AE in context restoration. In the restored text, GMSA-AE only has the last word inconsistent with the original text, *i.e.*, restoring "it" to its plural form "they". In contrast, TCP-AE not only exhibits inconsistencies in some word expressions (such as "medication" and "drugs") but also displays large segments of discrepancies with the original text.

Table 11: An example showing GMSA-AE and TCP-AE's context restoration performance. Text highlighted in yellow indicates discrepancies from the **Original Context**.

| Original Context | GMSA-AE | TCP-AE |
|---|---|---|
| Craig F. Walker \| Boston Globe \| Getty Images
The Trump administration is making good on its latest effort to lower out-of-pocket drug costs for Medicare recipients, but its approach could also limit drug options or even risk eliminating coverage of some prescriptions. The Centers for Medicare and Medicaid Services proposed late Monday changes to Medicare Advantage and Medicare Part D. Among the changes, it would allow insurers to stop covering certain "protected" classes of drugs used to treat common ailments like depression, cancer and HIV. When Congress added a prescription drug benefit to Medicare in 2003, it required insurers to cover at least two different drugs to treat any particular ailment. It also set aside six protected classes of medication where insurers were required to cover "all or substantially all" of the drugs offered to ensure seniors got whatever treatment they needed, like for cancer. The Trump administration thinks that gives drug manufacturers greater negotiating power on prices. The proposal is meant to give insurers more leverage and drive prices down by allowing them to drop coverage of certain drugs in a protected class if it | Craig F. Walker \| Boston Globe \| Getty Images
The Trump administration is making good on its latest effort to lower out-of-pocket drug costs for Medicare recipients, but its approach could also limit drug options or even risk eliminating coverage of some prescriptions. The Centers for Medicare and Medicaid Services proposed late Monday changes to Medicare Advantage and Medicare Part D. Among the changes, it would allow insurers to stop covering certain "protected" classes of drugs used to treat common ailments like depression, cancer and HIV. When Congress added a prescription drug benefit to Medicare in 2003, it required insurers to cover at least two different drugs to treat any particular ailment. It also set aside six protected classes of medication where insurers were required to cover "all or substantially all" of the drugs offered to ensure seniors got whatever treatment they needed, like for cancer. The Trump administration thinks that gives drug manufacturers greater negotiating power on prices. The proposal is meant to give insurers more leverage and drive prices down by allowing them to drop coverage of certain drugs in a protected class if they | Craig Walker \| Boston Globe \| Getty Images
The Trump administration is making good on its latest effort to lower out-of-pocket medication costs for Medicare recipients, but its approach could also limit drug options or even risk eliminating coverage of some prescriptions. The Centers for Medicare and Medicaid Services proposed late Monday changes to Medicare Advantage and Medicare Part D. Among the changes, it would allow insurers to stop covering certain "protected" drugs used to treat common ailments like depression, cancer and HIV. The Centers for Medicare and Medicaid Services proposed changes to Medicare Advantage and Medicare Part D. Among the changes, it would allow insurers to stop covering certain drugs that are used to treat common ailments like depression, cancer and HIV. The proposal would have added a prescription drug benefit to Medicare Part B, which currently covers only doctor visits and lab tests. Congress added the prescription drug benefit in 2003 to require insurers to cover at least two different drugs to treat any of the "essential drugs" offered to seniors, regardless of whether they were covered by Medicare |

## L  LANGUAGE MODEL USAGE STATEMENT

During the preparation of this manuscript, we utilize a large language model as a writing assistant. Its primary role is to refine and polish our paper, including the descriptions of our methodology and the presentation of mathematical derivations. This is done to improve the overall clarity, precision, and readability of the paper. All core ideas, experimental designs, and results are original work of the authors.

