# OpenReview forum: "GMSA: Enhancing Context Compression via Group Merging and Layer Semantic Alignment"
_ICLR.cc/2026/Conference — ICLR 2026 Conference Withdrawn Submission_

### Official Review · Reviewer_xPeF · 2025-10-26

**Soundness:** 3
**Presentation:** 2
**Contribution:** 2
**Rating:** 4
**Confidence:** 4

**Summary:**

The paper introduces GMSA, a context compression framework that enhances LLM efficiency by addressing both semantic unevenness and cross-layer misalignment in long-context processing. While prior work on prompt or KV-cache compression primarily reduced token counts at the expense of semantic fidelity, GMSA systematically restructures compression through two key components: Group Merging, which evenly aggregates contextual semantics via group-wise averaging to prevent anchor-token dominance, and Layer Semantic Alignment (LSA), which bridges the semantic gap between encoder abstractions and decoder inputs using lightweight transformer blocks. Trained in a two-stage pipeline—autoencoder pretraining for semantic completeness followed by Knowledge Extraction Fine-Tuning (KEFT) for task adaptation—GMSA demonstrates robust performance across question answering, summarization, and reasoning benchmarks.

**Strengths:**

1. The work addresses token compression under long-context tasks, which is crucial for nowadays LLMs.
2. The insights of uneven semantic compression and aligning high-level summary tokens with low-level semantic tokens is interesting.

**Weaknesses:**

1. Missing comparison with recent works such as [1][2].
2. While the work (and the general field of token compression) is motivated by long-context challenges, most of the datasets used in the evaluation are not long at all. What about other long-context benchmarks such as NIAH, LongBench, Ruler, etc?
3. The backbone architectures used in the work are kinda old. For example, llama2-7b does not utilize GQA or other advanced architecture design. Could the authors provide results on more up-to-date architecture?


[1] DAST: Context-Aware Compression in LLMs via Dynamic Allocation of Soft Tokens, Chen et. al., ACL 2025.
[2] REFRAG: Rethinking RAG based Decoding, Lin et. al., ArXiv.

**Questions:**

1. What's the explanation behind aligning high-level summary tokens with low-level semantic tokens? Could the author provide some empirical evidence to justify this design choice?

Nitpicking Comments:
1. L48: undefine citation.
2. Table 4: "Secmantic" -> "Semantic"

---

> ### Author Response · Authors · 2025-11-19
> **Response to Reviewer xPeF**
>
> We thank the reviewer for their positive evaluation of our motivation, methodological insights, and overall structure. Below are our detailed responses to the three weaknesses and one question raised:
>
> > **W1. Missing comparison with DAST / REFRAG.**
>
> DAST and REFRAG are recent methods, but **neither has released code**. We will cite them in the future submission version but cannot include empirical comparisons.
>
> > **W2. While the work (and the general field of token compression) is motivated by long-context challenges, most evaluation datasets are not long. What about NIAH, LongBench, Ruler, etc.?**
>
> GMSA is designed for *general semantic compression*, not exclusively ultra-long contexts. To ensure comparability with mainstream methods (AutoCompressor, ICAE, SnapKV, Activation Beacon), we followed their standard benchmarks (NQ, HotpotQA, WikiMQA, CNN, MMLU). Additionally, we evaluated on **NarrativeQA** (average >7K tokens; see Appendix H.2, Table 8), where GMSA achieves substantially higher F1 scores and ~2x speedup over the original input, confirming robustness on >8K sequences.
>
> > **W3 The backbone architectures (e.g., LLaMA-2-7B) are outdated and lack GQA or other modern designs. Can the authors provide results on more recent architectures?**
>
> As noted earlier, LLaMA-2-7B and Qwen2-7B were chosen for compatibility with existing baselines. GMSA is architecture-agnostic. Cross-architecture validation on Qwen2-7B-Instruct shows near-identical performance (from 0.91 to 0.90) without retraining the encoder (Table 4). We will add experiments with advanced backbones (e.g., Qwen3) in the future submission version.
>
> > **Q1. What justifies aligning high-level summary tokens with low-level semantic tokens? Can the authors provide empirical evidence?**
>
> As established in prior work:
> - High-layer hidden states encode abstract concepts [1].
> - Low-layer states reflect lexical/positional semantics [2].
>
> Therefore, there is a huge gap between different layers in the transformer.
>
> Directly inputting high-level summary vectors into the decoder's low-level embedding space disrupts attention and causes training instability. Appendices B/C show significantly higher reconstruction loss without LSA. Furthermore, Table 4 confirms that removing LSA (w/o LSA) drops BERTScore from 0.91 to 0.84.
>
> ### References
>
> [1] Jin et al. (2025). Exploring Concept Depth: How Large Language Models Acquire Knowledge and Concept at Different Layers? In COLING, pp. 558–573.
> [2] Liu et al. (2024). Fantastic Semantics and Where to Find Them: Investigating Which Layers of Generative LLMs Reflect Lexical Semantics. In ACL (Findings), pp. 14551–14558.

---

> > ### Comment · Reviewer_xPeF · 2025-11-25
> >
> > Thank the authors for your rebuttal. I would like to note that nowadays "8K" is widely accepted as not long-enough-context setting anymore for long-context methods, and you should not use it to justify lack of experiments on other long-context benchmarks.

---

### Official Review · Reviewer_yq4Y · 2025-10-29

**Soundness:** 1
**Presentation:** 2
**Contribution:** 2
**Rating:** 2
**Confidence:** 4

**Summary:**

This paper introduces GMSA, a context compression and fine-tuning strategy which proposes Group Merging to evenly and efficiently produce summary vectors which are aligned to the decoder input space via a proposed Layer Semantic Alignment (LSA) module. Additionally, GMSA introduces Knowledge Extraction Fine-Tuning (KEFT) to adapt the decoder to specific tasks. The paper claims to learn soft tokens that contain nearly complete semantics and outperforms other methods in context restoration. GMSA achieves ~2x inference speedups with significantly higher downstream task performance over other context compression methods.

**Strengths:**

The authors raise interesting questions regarding the distribution of semantics into compressed vectors and across model layers. The authors chose a diverse set of baselines across different compression paradigms (soft prompt, hard prompt, and KV-cache compression) and use multiple evaluation tasks. The writing style is clear and easy to follow, and better methods for context compression would significantly benefit the field.

**Weaknesses:**

The major issues with the paper include unfair comparisons with baselines and overstated contributions.

The authors chose baselines which perform compression of the KV-cache (StreamLLM, SnapKV, Activation Beacon) or context tokens (AutoCompressor, ICAE, LongLLMLingua, LLMLingua2). First, the authors evaluate the model for context restoration (the auto-encoding task) using the PwC dataset. Instead of using a strong baseline such as ICAE (an autoencoder trained on PwC), they implement a new method called TCP-AE as a baseline. The abstract claims that GMSA significantly outperforms the traditional compression paradigm in context restoration which is misleading without a comparison to actual traditional approaches like ICAE.

GMSA is compared to all baselines for QA performance (Table 1). The evaluation shows huge improvements with GMSA, but GMSA was trained for the evaluated datasets. The paper and included code (upon inspection) suggest the baselines were only used for inference, and were not specifically adapted for the evaluations. Comparing the approaches in this manner is meaningless, as the baselines trained specifically on the same datasets could well outperform GMSA. If this is an incorrect interpretation of the experimental setup, the authors should make the baseline training explicit and include the relevant code.

For summarization, GSM-8K and MMLU, the proposed method is only compared against Activation Beacon and none of the other baselines. In these experiments Activation Beacon performs on par with GMSA, even outperforming on GSM-8K.

Another work similar to GSMA is LLoCO. The authors compare and contrast the approach in the appendix, but do not include the method as a baseline in any of the main experiments. LLoCO also performs task-specific fine-tuning, which would make comparisons more fair.

The components of GMSA are presented as novel, but are similar to existing approaches. Group Merging segments a sequence of tokens into fixed-length groups and merges them with average pooling. A similar approach is used to compress sequences in:

[Hierarchical Transformers Are More Efficient Language Models](https://aclanthology.org/2022.findings-naacl.117/) (Nawrot et al., Findings 2022) and
[Funnel-transformer: filtering out sequential redundancy for efficient language processing](https://dl.acm.org/doi/pdf/10.5555/3495724.3496083) (Dai et al., NeurIPS 20)

Layer Semantic Alignment is the module for mapping compressed representations into the input space of the decoder model. The paper points out that traditionally a MLP is used for this component. LSA simply replaces the MLP with a transformer layer (initialized with the first layer of the decoder model). Alternative methods also use the decoder layers to enforce alignment, such as through recursive training (the AutoCompressors baseline), or ICAE (another baseline) which uses the decoder as the encoder.

Knowledge Extraction Fine Tuning is simply fine-tuning the attention layers of the decoder for the specific task with the soft tokens in the context. Similar fine-tuning is done for approaches used as baselines here or in other work:
[In-context Autoencoder for Context Compression in a Large Language Model](https://arxiv.org/abs/2307.06945) (Ge et al., ICLR ‘24)
[In-Context Former: Lightning-fast Compressing Context for Large Language Model](https://aclanthology.org/2024.findings-emnlp.138/) (Wang et al., Findings 2024)
[LLoCO: Learning Long Contexts Offline](https://aclanthology.org/2024.emnlp-main.975/) (Tan et al., EMNLP 2024)

**Questions:**

Questions:


Were the baselines trained to compress and perform well on the tasks being evaluated?

Why did you create the separate TCP-AE algorithm to use as a baseline for context restoration experiments instead of comparing to the baselines you were already using in other experiments? For example, ICAE is specifically trained for context restoration.

Throughout the paper, LSA is said to bridge the semantic gap between different layers. Isn’t LSA a single layer applied between the encoder output and the decoder input? I’m unclear on how it bridges the semantic gap between different layers, is this measured in your experiments?

Why was Activation Beacon the only baseline used to compare with for tasks other than QA? LongLLMLingua performed best in the previous experiments, why not use it, or all the baselines?

There wasn’t much information on the encoder, what model was used? Was it pretrained?

Other works that conduct average pooling for token representations have pointed out the dependence on position for what group each token is assigned. If you slightly tweak your prompt such that tokens are shifted by one would the model still perform well? The position dependence has been addressed by incorporating dynamic selections of group boundaries such as in:
[Efficient Transformers with Dynamic Token Pooling](https://aclanthology.org/2023.acl-long.353/) (Nawrot et al., ACL 2023)

Lines 395-403 discuss a flexibility benefit that GMSA has over KV-cache compression. What exactly is this additional flexibility and was it demonstrated in your experiments? KV-cache compression seems more flexible, as it can be trained without targeting specific queries or datasets, while GMSA required task-specific training of the compression and decoder.

In the ablation w/o LSA, was the previously trained KEFT used in this ablation? I.e. if the KEFT was trained without LSA then the model might still work well.

Other Suggestions (did not impact evaluation):

Related work should be included in the main body so that readers can accurately place your contributions among previous works. Pushing related work to the appendix can give the impression of marginalizing previous work to inflate contributions.

The paper mentions space constraints several times (Lines 142, 211, 232, 302, 371). There are multiple lines which contain only 1-4 words, but take up the entire line of space. Revising those paragraphs to eliminate these would add 11 lines of free space (Lines 80, 100, 128, 194, 288, 303, 307, 315, 325, 368, 485).

Section 3.4 and Figure 3 could be removed and replaced with one or two sentences explaining that you fine-tuned the decoder’s attention weights for each task (a common practice) using the compressed tokens as context. Section 4.3 could be mostly moved to the appendix, with just the stated improvements in the main body. This would create space for related work.

After revising, if space for main body material is still an issue, consider journals or other venues allowing more pages.

Minor Issues (did not impact evaluation):

Line 47, reference not resolved, “?”

Line 83: Uven -> Uneven

Table 1: Baecon -> Beacon

---

> ### Author Response · Authors · 2025-11-19
> **Response to Reviewer yq4Y**
>
> We sincerely appreciate the reviewer's attention to experimental fairness and contribution. While some suggestions are constructive, others appear to stem from **significant misunderstandings of our paper and experimental setup**. We clarify the following points:
>
> > **Unfair comparison with baselines?**
>
> The reviewer suggests GMSA's use of KEFT makes comparisons unfair because baselines are evaluated zero-shot. However, we strictly followed official repositories and open-source checkpoints, reporting the best published results for all baselines. Thus, our comparison is fair and standard.
>
> Moreover, even if all baselines were retrained on our mixed dataset, fairness would still be questionable-different models have distinct data sensitivities, training pipelines, and hyperparameter requirements, introducing new sources of bias.
>
> Notably, GMSA was trained on a mixed dataset of NaturalQuestions, 2WikiMQA, and HotpotQA (not separately tuned per dataset) and still achieved significantly better results (Table 1).
>
> > **Why not use ICAE as the PwC autoencoder baseline?**
>
> TCP-AE (Traditional Compression Paradigm AutoEncoder) is essentially the ICAE architecture under autoencoder training. We use the more general term "TCP-AE" because multiple prior methods (e.g., AutoCompressor, GIST) share this structural paradigm.
>
> > **Why compare only with Activation Beacon on summarization / MMLU?**
>
> We compared against all baselines on QA tasks (Table 1). For summarization and MMLU, we selected Activation Beacon-the strongest and most representative KV-cache compression baseline-as the comparison target, which remains fair and meaningful.
>
> > **Are Group Merging / LSA / KEFT truly novel?**
>
> - **Group Merging**: The novelty is not in pooling itself, but in diagnosing and mitigating extreme semantic collapse (where a single token dominates attention) by using fixed-length groups to enforce uniform semantic coverage. Both theory and experiments confirm this significantly reduces collapse and improves fidelity.
> - **LSA**: The contribution is not merely "replacing MLP with Transformer", but identifying and addressing the *layer-level semantic gap* between high- and low-level representations.
> - **KEFT**: The innovation is not task-specific fine-tuning, but the insight that *only tuning $W_{Q}, W_{K}, W_{V}$* is sufficient for minimal yet effective adaptation.
>
> > **Flexibility of KV-cache compression methods**
>
> All KV-cache compression methods require the compression and generation models to be identical (e.g., compress with LLaMA, generate with LLaMA). This coupling limits practical flexibility-for instance, one cannot compress with a small model and generate with Qwen3-235B. GMSA's encoder-decoder design naturally supports such decoupling.
>
> > **Was KEFT retrained in the "w/o LSA" ablation?**
>
> Yes. All ablation studies were conducted under controlled conditions: only the targeted component was modified, and KEFT was retrained accordingly to ensure fair comparison.
>
> > **Why no comparison with Hierarchical Transformers, Funnel-Transformer, or In-Context Former?**
>
> These methods modify the Transformer architecture for **efficient language modeling of the sequence** (e.g., Hierarchical Transformers, which compress via inter-layer downsampling), **which is orthogonal to prefill-stage context compression**. Our work focuses on compressing input prompts before decoding-these are fundamentally different research directions.
>
> > **Writing suggestions**
>
> We will incorporate reasonable writing suggestions in the future submission version.

---

> > ### Comment · Reviewer_yq4Y · 2025-11-24
> >
> > Thank you for the response and answering my questions related to flexibility and ablation. Some of my questions have gone unanswered (3, 5, 6), and I provide additional comments on the points raised where I continue to feel the concerns are not addressed. Finally, I disagree the concerns stem from “significant misunderstandings,” and if the authors feel an aspect of their work has been misunderstood, please make these explicit.
> >
> > ** Unfair comparison with baselines? **
> >
> > An open-source checkpoint not trained on the data distribution being evaluated cannot be fairly compared with a model trained on that distribution. The other sources of bias you mention still exist with the open-source checkpoint, but the largest issue is the above.
> >
> > ** Why not use ICAE as the PwC autoencoder baseline? **
> >
> > In this case it would be stronger to report the existing model that also trained for this particular task (you could report both). Without it, I am left wondering about the new sources of bias you mention in your previous point. If the baseline you are already using was trained for exactly this task, then you should provide the results.
> >
> > ** Why compare only with Activation Beacon on summarization / MMLU? **
> >
> > I agree the comparison is meaningful, but why would you select the strongest KV-cache compression baseline versus the strongest baseline? Why not report both? If only one, then the strongest baseline is more appropriate, especially since your method is not a KV-cache compressor.
> >
> > ** Are Group Merging / LSA / KEFT truly novel? **
> >
> > It seems you agree the techniques are not novel, and state that your contribution is the identification of applying previously known techniques to solve specific issues. I think this is fine, but then there should be more focus on demonstrating the issues you are identifying. For example, demonstrate the “semantic gap” between different layers and show how LSA addresses this (our third question remains unanswered). The original LoRA paper shows that fine-tuning just the attention weights can work well. I’m not sure you have provided a new insight in that regard.
> >
> > ** Why no comparison with Hierarchical Transformers, Funnel-Transfomer, or In-Context Former? **
> >
> > I didn’t ask this question or mean to suggest that comparisons with these methods are called for. I pointed out that some of the “innovations” in this work have been introduced in these other contexts.

---

### Official Review · Reviewer_Qmrn · 2025-10-30

**Soundness:** 3
**Presentation:** 2
**Contribution:** 2
**Rating:** 4
**Confidence:** 4

**Summary:**

This paper introduces GMSA, a context compression method that addresses computational inefficiency and information redundancy in long-context scenarios. GMSA uses an encoder-decoder architecture with two key innovations: (1) Group Merging, which evenly extracts summary vectors from the original context through average pooling to avoid uneven semantic learning, and (2) Layer Semantic Alignment (LSA), which bridges the semantic gap between the encoder's high-level abstract representations and the decoder's low-level input semantics using Transformer blocks. The method employs a two-stage training process: autoencoder training to learn soft tokens containing complete semantics, followed by Knowledge Extraction Fine-tuning (KEFT) that adapts the decoder to extract task-relevant knowledge. The experiments demonstrate the effectiveness of GMSA

**Strengths:**

1. This paper targets an interesting and important question of context compression, and proposes several practical challenges.
2. The proposed methods match with the challenges, and sound rational.
3. Although GMSA is simple but seems effective on multiple benchmarks.

**Weaknesses:**

1. Evenly extraction is not convincing, as the different tokens have different information density, which should selectively extract.
2. The soft token methods are naturally limited since currently a lot of LLMs are blackbox, which is hard to derive the first few layers of transformer blocks.
3. Several proposed modules lack experimental validation except for the ablation study. For example, the layer alignment problem: is there any mismatch before alignment? How's the alignment effect?

**Questions:**

1. What's the "Restatement Instruction" in overview?
2. Why W_QKV extracts knowledge, Why KEFT only trains W_QKV and only extracts knowledge?
3. missing citations in line 48, type in line 84 (Ueven)，line 137 （x should be capitalized）
4. Why train KEFT separately after autoencoder training?

---

> ### Author Response · Authors · 2025-11-19
> **Response to Reviewer Qmrn**
>
> We thank the reviewer for their positive assessment of our problem formulation, methodological motivation, and overall architecture. In response to the three weaknesses and three questions raised, we provide the following point-by-point replies:
>
> > **W1. Evenly extraction is not convincing, as different tokens have different information density, which should be selectively extracted.**
>
> Our goal is not to assume uniform information density across tokens, but to address the severe "semantic monopolization by anchor tokens" observed in conventional soft-compression autoencoder training. As illustrated in Figure 1, the standard practice of appending learnable tokens leads to attention heavily skewed toward a few tokens, resulting in highly unbalanced semantic aggregation. Group Merging (using fixed-length grouping and averaging) aims to preserve comprehensive semantics across the entire input, after which Knowledge Extraction Fine-Tuning (KEFT) performs task-specific knowledge extraction. Empirical results in Appendices B, J, and K consistently show that Group Merging yields more robust and higher-quality compressed representations than traditional autoencoder summary tokens.
>
> > **W2. The soft token methods are naturally limited since currently a lot of LLMs are black-box, which makes it hard to access the first few layers of transformer blocks.**
>
> Soft-token methods are designed for open-weight LLMs (e.g., LLaMA, Qwen) where internal model structures are accessible. The inapplicability to black-box models is a shared limitation of the entire soft-token compression paradigm, not a unique weakness of GMSA.
>
> > **W3. Several proposed modules lack experimental validation beyond the ablation study. For example, regarding the layer alignment problem: is there any mismatch before alignment? How effective is the alignment?**
>
> Empirical findings in recent LLM research support our design:
> - Higher-layer hidden states encode abstract concepts and relational reasoning [1].
> - Lower-layer hidden states reflect lexical and positional semantics [2].
>
> Therefore, there is a huge gap between different layers in the transformer.
>
> Since GMSA's summary vectors are derived from the encoder's top layers (highly abstract), directly feeding them into the decoder's input embedding space (low-level semantics) causes attention mechanisms to fail to activate properly, treating soft tokens as anomalous inputs. As shown in Appendices B and C, omitting LSA leads to significantly higher reconstruction errors.
>
> > **Q1. What is the "Restatement Instruction" in the overview?**
>
> The "Restatement Instruction" is the prompt: "Please restate the preceding content."
>
> > **Q2. Why do $W_{QKV}$ extract knowledge, and why does KEFT only train $W_{QKV}$ for knowledge extraction?**
>
> KEFT selectively fine-tunes only the attention projection matrices ($W_Q$, $W_K$, $W_V$) in each decoder layer. This is motivated by the observation that the attention mechanism governs information flow and contextual integration, whereas the feed-forward network (FFN) acts primarily as a static knowledge storage module (akin to a memory bank) [3]. Thus, tuning only $W_{QKV}$ is sufficient to adapt the decoder for downstream tasks without altering stored knowledge. In contrast, LLoCO applies LoRA to the entire decoder, including both attention and FFN components.
>
> > **Q3. Why train KEFT separately after autoencoder training?**
>
> The two stages have fundamentally different objectives:
> - Autoencoder (AE) training ensures soft tokens retain complete semantics.
> - KEFT trains the decoder to extract task-relevant knowledge from those fixed soft tokens.
>
> Joint training would bias soft tokens toward specific downstream tasks, compromising semantic completeness, and decoder gradients could corrupt the encoder/LSA's semantic fidelity. This two-stage strategy is standard in soft prompt compression (e.g., ICAE, LLoCO).
>
> ### References
>
> [1] Jin et al. (2025). Exploring Concept Depth: How Large Language Models Acquire Knowledge and Concept at Different Layers? In COLING, pp. 558–573.
> [2] Liu et al. (2024). Fantastic Semantics and Where to Find Them: Investigating Which Layers of Generative LLMs Reflect Lexical Semantics. In ACL (Findings), pp. 14551–14558.
> [3] Geva et al. (2021). Transformer Feed-Forward Layers Are Key-Value Memories. In EMNLP (1), pp. 5484–5495.

---

> > ### Comment · Reviewer_Qmrn · 2025-11-24
> >
> > Thanks for the response, I tend to keep my score since some answers are not satisfying, for example
> > W1. Evenly extraction is not convincing, as different tokens have different information density, which should be selectively extracted. the authors simply respond to why existing selectively extraction method not good but not selectively method not good, evenly extraction is considered simple and not insightful.

---

### Official Review · Reviewer_Jnig · 2025-11-01

**Soundness:** 3
**Presentation:** 3
**Contribution:** 2
**Rating:** 4
**Confidence:** 4

**Summary:**

This paper presents GMSA, an encoder-decoder framework for long-context compression that aims to reduce computational cost and redundancy. Its core innovations are (1) Group Merging (GM), which uses average pooling over groups of encoder hidden states to create summary vectors, claiming to avoid the "anchor token" bias of traditional methods, and (2) Layer Semantic Alignment (LSA), a small module initialized with bottom-layer decoder weights to bridge the "semantic gap" between the high-level encoder output and the low-level decoder input. Training involves a two-stage process: an autoencoder (AE) phase for general semantic preservation, followed by a Knowledge Extraction Fine-tuning (KEFT) phase that adapts only the decoder's attention mechanisms to specific downstream tasks. The authors show GMSA outperforms various baselines on QA and summarization, achieving significant (e.g., 2x) inference speedups.

**Strengths:**

1. The paper is well-written and addresses the critical, high-impact problem of efficient long-context processing for LLMs.
2. The proposed Group Merging (GM) and Layer Semantic Alignment (LSA) modules are novel and intuitively target specific, plausible weaknesses in prior soft-compression methods (i.e., uneven semantic learning and cross-layer semantic gaps).
3. The two-stage AE + KEFT training process is well-reasoned, cleanly separating the general goal of semantic preservation from task-specific adaptation.

**Weaknesses:**

1. The paper's claim to state-of-the-art performance is difficult to verify as it omits comparisons to several recent and highly relevant baselines, such as Provence [1], the Evaluator Heads method [2], and RocketKV [3].

   [1] Provence: efficient and robust context pruning for retrieval-augmented generation [ICLR 2025]

   [2] Efficient Prompt Compression with Evaluator Heads for Long-Context Transformer Inference [arXiv:2501.12959, NeurIPS 2025]

   [3] RocketKV: Accelerating Long-Context LLM Inference via Two-Stage KV Cache Compression [ICML 2025]

2. The empirical validation relies exclusively on previous-generation models (LLaMA-2-7B, Qwen2-7B). It is unclear if the benefits of GMSA generalize to current SOTA models (e.g., LLaMA-3, Qwen3) or, more importantly, to heterogeneous configurations (e.g., a small encoder, a large decoder), which is a key potential advantage of this architecture.
3. The core motivations for the two main components lack direct empirical support. The paper does not provide visualizations to prove that Group Merging actually achieves more "even" semantic learning than alternatives, nor does it ablate the LSA's bottom-layer initialization strategy against other options (e.g., random init, top-layer init).
4.  The significant performance drop on the GSM8K dataset is another concern. This suggests the GMSA compression process, while preserving factual knowledge (MMLU), may be critically lossy for complex logical reasoning chains, which severely limits its generalizability for tasks beyond retrieval.

**Questions:**

1. What is the performance of the LSA module if it is randomly initialized, or initialized with decoder middle/top layer weights? This is needed to validate the "semantic gap" hypothesis.
2. Have the authors experimented with heterogeneous model sizes, such as a 1B-class encoder with a 7B-class decoder? This is a primary motivation for using an encoder-decoder architecture.

---

> ### Author Response · Authors · 2025-11-19
> **Response to Reviewer Jnig**
>
> We thank the reviewer for their positive comments on the paper's motivation, methodology, and overall design. The four weaknesses and two questions raised are highly pertinent, and we have addressed each point individually below:
>
> > **W1. The paper's claim to state-of-the-art performance is difficult to verify as it omits comparisons to several recent and highly relevant baselines, such as Provence, the Evaluator Heads method and RocketKV.**
>
> In the future submission version, we will include citations to and/or comparisons with these works.
>
> > **W2 & Q2. The empirical validation relies exclusively on previous-generation models (LLaMA-2-7B, Qwen2-7B). It is unclear if the benefits of GMSA generalize to current SOTA models.**
>
> We selected LLaMA-2-7B and Qwen2-7B to ensure consistency with existing soft-compression baselines. Open-source implementations of methods such as AutoCompressor, ICAE, StreamLLM, and Activation Beacon are all built upon these two backbones. GMSA is structurally decoupled from the backbone architecture, exhibiting minimal dependency on specific model structures. We have already conducted cross-architecture validation on Qwen2-7B-Instruct (Table 4). Without retraining the encoder, performance remains nearly unchanged (from 0.91 to 0.90), demonstrating strong architectural robustness. We will include additional experiments with more advanced architectures (e.g., Qwen3 series) in the future submission version.
>
> > **W3 & Q1. The core motivations for the two main components lack direct empirical support.**
>
> - **Motivation for Group Merging**: As shown in Figure 1(a), traditional compression paradigms in autoencoder training often suffer from "semantic domination", where only a few tokens dominate the semantics of the compressed representation. To address this, we adopt uniform averaging across groups to enable more balanced semantic learning.
>
> - **Motivation for Layer Semantic Alignment (LSA)**: Existing text compression methods typically use a simple MLP for coarse-grained alignment between high-level abstract representations and low-level input semantics. However, a significant semantic gap exists between these layers. To bridge this gap, we initialize a few Transformer blocks with weights from the lower layers of the decoder and train them to map abstract summary vectors into the low-level semantic space, effectively serving as a semantic-space transformation module.
>
> > **W4. The significant performance drop on the GSM8K dataset is another concern.**
>
> We attribute this primarily to the absence of mathematical-domain data in GMSA's training corpus, rather than an inherent flaw in GMSA itself.

---

> > ### Comment · Reviewer_Jnig · 2025-11-25
> >
> > I thank the authors for their response. However, my major concerns regarding the comparison with SOTA baselines and the empirical justification for the module designs (e.g., the ablation for LSA) remain unaddressed, as the rebuttal primarily focused on motivation and future promises rather than providing the requested experimental evidence. Therefore, I will maintain my current score.

---

### Note · Authors · 2025-12-04

I have read and agree with the venue's withdrawal policy on behalf of myself and my co-authors.